# Post-whaling shift in mating tactics in male humpback whales

Rebecca Dunlop [1✉] & Celine Frere[1]

Recent studies have shown behavioural plasticity in mating strategies can increase a population's ability to cope with anthropogenic impacts. The eastern Australian humpback whale population was whaled almost to extinction in the 1960s (~200 whales) and has recovered to pre-whaling numbers (>20,000 whales). Using an 18-year dataset, where the population increased from approximately 3,700 to 27,000 whales, we found that as male density increased over time, the use of mating tactics shifted towards more males engaging in non-singing physical competition over singing. Singing was the more successful tactic in earlier post-whaling years whereas non-singing behaviour was the more successful tactic in later years. Together, our study uncovers how changes in both local, and population-level male density resulted in a shift in the frequency, and fitness pay-off, of alternative mating tactics in a wild animal. This individual-level plasticity in male humpback whale mating tactics likely contributed to minimising their risk of extinction following a dramatic change in their social landscape due to whaling.

[1] School of Biological Sciences, University of Queensland, St Lucia, 4072 Brisbane, QLD, Australia. ✉email: r.dunlop@uq.edu.au

Males within a population may use alternative mating tactics, such as mate guarding, surreptitious sneaker, territoriality, fighting, and displaying to gain access to a female[1–4]. According to the fitness payoff (reproductive success), individuals have a set of rules specifying when it should use each tactic[5]. If these tactics form a conditional strategy, the likelihood each one is expressed depends on variations in extrinsic and intrinsic factors[3,6,7]. One such factor is conspecific density. For example, local variations in female density influences the mating behaviour of male ungulates, from holding dispersed territories or female-following in low female density, to forming leks in high female density[8,9]. Male elephants may switch from guarding to bull tactics depending on the number of available females[10]. Local male density has been found to influence courtship tactics in fish (e.g., squaretail grouper, *Plectropomus areolatus*[11]) and amphibians (e.g., Woodhouse's toad, *Bufo woodhousii*[12]). In addition, population-level density may also change the prevailing tactics of males. For example, following a severe decline in the pronghorn (*Antilocapra americana*) population of the National Bison Range, males switched from using a territorial defence tactic, to holding a harem[13]. We can therefore assess the influence of conspecific density on individual male mating tactics at a local level, and a population level. While a few experimental studies have demonstrated that the frequency and fitness pay-off of alternative mating tactics can be influenced by male density at a population level, empirical studies in wild animals are scarce given the requirement for a severe change in population density.

Whaling has severely impacted multiple large whale population since the last few hundred years culminating in approximately 1.8 million large whales being killed in the Southern Ocean in the 20th century[14]. The whaling moratorium has resulted in the recovery of many humpback whale populations (e.g., the North Atlantic population[15], South Atlantic population[16], and Southern Ocean populations[17]). For example, the eastern Australian humpback whales (*Megaptera novaeangliae*), which feeds in the Southern Ocean, was hunted almost to extinction. Numbers reduced from approximately 26,000 pre whaling, to approximately 200 in the 1950s and 1960s[18,19]. The cessation of whaling led to an unprecedented recovery in which the population number has steadily increased by 11% per year[20]. Given whaling targeted mature adult whales, and assuming humpback whales live for approximately 60 years, only now (in the 2020 s) is this population likely to include animals that have lived their entire lifespan. Such a dramatic change in their social landscape could have meant their pre-whaling breeding system ceased to function and the population could have died out. Obviously, this was not the case given the rapid population recovery.

Within breeding aggregations, the operational sex-ratio is skewed towards males[21]. This results in high levels of conflict when competing for access to females. Humpback whale males express two alternative, and likely conditional, mating tactics. The first is physical competition. Male humpback whales, during the breeding season, will temporarily join and escort a female and attempt to breed with her[22,23]. This male-female group can sometimes attract additional males and form a 'competitive group'. Here, males physically fight, jostle, and compete for the primary escort position[21,24–26] with larger, sexually mature males tending to occupy the primary escort position[27]. Ultimately, this can lead to the displacement of the original escort from the primary escort position. The second is singing. Though there is much debate in the literature as to the function of song, it is generally accepted that song functions to attract females, male-male competition, facilitate a male lekking-type system, or a combination of these functions[28]. Singers are sexually mature[27] males[24,29,30]. They are often found displaying alone[26,31], then may join a female whilst singing to become a primary escort[30,32]. Like non-singing

males, these groups can attract more males to form a competitive group[32,33], which can then lead to physical fighting and escort displacement. They are sometimes joined by non-singing males to form an adult pair[29–31], though it is still unclear whether this is a competitive interaction related to social ordering, or a cooperative interaction. In the breeding grounds, and on migration, group membership is dynamic, with males constantly moving in and out of each other's immediate social environment. This results in continuous temporal variations in local male density. As the presence of other males has been shown to influence whether migrating male humpbacks began, or ceased, singing[32,34], male humpback whale mating tactics are likely to be conditionally expressed according to male density. This assumption of behavioural plasticity will be tested further in the current study.

Here, we use more than a decade long dataset on humpback whale behaviour to investigate the consequences of large changes in population density on the frequency, and relative pay-off, of male alternative mating tactics (singing or not singing and sometimes physically competing) in migrating humpback whales. Relative success of each tactic, at an individual level, was quantified as the male joining a female. This does not assume a successful mating took place, rather, that the male had a greater chance of mating with a female compared to those that were alone. When this male-female group attracted additional males, this was assumed to be a cost, given there was a chance that the original male would be displaced from his position as primary escort thus losing its exclusive access to the female and its chance to mate. We first quantified the proximate effect of local male density on the likelihood males would express one tactic over the other, i.e., evidence of individual-level behavioural plasticity. Earlier studies carried out in this field site found that singing males were less likely to sing in the presence of other competing males[32,34]. Here, we extended this finding to include non-singing males. Next, accounting for this proximate effect, the ratio of alternative mating tactics with the population was tracked over time to determine if there was any evidence that males were shifting their mating strategy as the population increased post-whaling. Over the time of this study, the humpback whale population increased from approximately 3700 whales, to over 27,000. Finally, a measure of the pay-off of each tactic was developed and tracked over time to determine if the increasing population changed the relative success of each tactic. The results present an empirical study of the influence of increasing male density on the frequency, and fitness pay-off, of alternative mating tactics in a wild animal.

## Results

### The proximate effect of male density on individual mating tactics.
For this analysis we chose one time period, the 2003/2004 dataset, which had the most instances of individual males being observed both singing and not singing. In addition, as the whale density increased post 2004, it became increasing difficult to focally follow individual whales without confusing them with other whales. In 2003/2004, whales were migrating through the study area at sufficiently low density to avoid confusion.

First, 86 unaccompanied singing males were extracted from the 2003/2004 dataset, and their location within the study area was recorded at the start of a song cycle using the acoustic array. Next, 31 unaccompanied non-singing males were extracted. They were assumed to be male given they were observed to actively approach and join a group (n = 31) and the fact that some of them later started to sing. Sexing results from a biopsy study (supplementary results) suggests that whales that actively join groups, and those that sing, are male. For these non-singing joining animals, visual observations were backtracked for 10 to 15 min (two or three

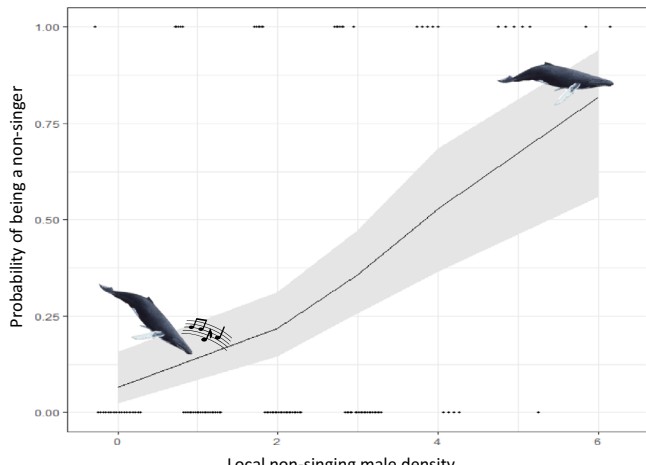

**Fig. 1 The probability of a lone focal whale not singing when first observed as a function of local non-singing male density.** The line is the model estimate of the relationship between the likely of first observing a male humpback whale singing or not singing and the local male density at the time measured as the number of other presumed males within a 5 km radius of the focal male. The model assumed a generalized linear binomial distribution, ($P = 0.0002$) and includes the 95% confidence interval (grey area). Raw datapoints are also included at the bottom (focal whales are singers; $n = 86$) and top (focal whales are non-singers; $N = 31$). Icons illustrate the likelihood of singing and non-singing behaviour.

surfacings) until they were sighted alone. To generate a measure of local male density, the number of other singing and non-singing (presumed) males were then counted within their social circle (within a 5 km radius), as outlined in the methods.

When first observed, lone focal males had between zero and six other non-singing presumed males (see supplementary methods and supplementary results for allocation of sex) and zero or one other singing whale within their social circle. In higher male densities, lone focal males were significantly more likely to be a non-singer (Fig. 1; Generalized linear model (GLM); $N = 117$; $Z = 3.697$, $P = 0.0002$). Those that were observed singing had three or less other non-singing males within their social circle, those that were not singing had four or more (Fig. 1). The presence of another singing male within their social circle had no significant effect on whether they were a singer or non-singer (GLM; $N = 117$; $Z = 0.484$, $P = 0.6274$).

The above analysis assessed the local environment of individual males at one timepoint: males observed alone and singing compared to males observed alone and not singing. However, for male breeding behaviour to be considered as a behaviourally plastic conditional strategy, the likelihood each tactic is expressed should depend on variations in intrinsic or extrinsic factors. To test this, focal males that were observed to switch between the two tactics ($N = 40$) were further analysed to determine if changes in local male density may have caused the switch. Lone singing males remained in the same area and did not make progress in any direction, suggesting song was being used as a broadcast signal. If joining a group, they made directional, intentional progress towards the group, to then join and match the group's swimming speed and direction. Most focal males that were first observed singing, then switched to non-singing behaviour at some point after joining another lone animal or female-calf pair ($N = 28$). Those that were joined by another singing male ($N = 1$), or another male ($N = 5$), ceased singing as soon as they were joined. The male pair then quickly split and both animals moved separately out of the study site. The rest ceased to sing whilst still alone ($N = 6$), then moved southwards to leave the

study site. For all males, their social circle was compared when first observed singing, to as soon as they ceased to sing.

We found when males switched from singing to non-singing behaviour, the number of other non-singing males (Fig. 2; Generalized Linear Mixed Model (GLMM); $N = 40$; $Z = 2.87$, $P = 0.004$) and singing males (Fig. 2; GLMM; $Z = 2.95$, $P = 0.003$) within their social circle were significantly higher. There was also a significant interaction between the two effects (Fig. 2; GLMM; $Z = -2.75$, $P = 0.006$). When singing, focal whales had between zero and three other non-singing males, and (usually) no other singers, within their social circle. When switching to non-singing behaviour, they usually had more than two other non-singing males (including any males within the joined group) and one other singer within their social circle. In addition, males were highly likely to switch from singing to non-singing behaviour in the presence of another singing male, even with a low number of other non-singing males within their social circle (Fig. 2). In other words, individual mating tactics were dependent on both the number, and tactics, of the other males within their social circle.

To summarise, these results and interactions suggest male humpbacks will choose to sing when the local male density is low, and cease singing when local male density increases and/or another whale begins to sing. The proximate effect of local male density was therefore accounted for in subsequent analyses by including the number of migrating males per day as covariate.

**Frequency of alternative mating tactics**. Here, a population-level approach was taken to determine if males were more likely to choose one tactic over the other as the population increased post-whaling from approximately 3700 in 1997 to over 27,000 in 2015. To do this, the number of singing and non-singing males per day were quantified and compared over time (1997 to 2015; $N = 123$ days) as outlined in the methods and supplementary results. In all datasets (1997, $N = 28$; 2003/2004, $N = 61$; 2008, $N = 14$ and 2014/2015, $N = 20$), the number of observed non-singing males per day always outweighed the number of singing males per day in that males would continually move through the study area throughout the day, with most of them not singing at any one time.

There was no evidence that singing whales migrated further from shore with increased population density. Singer positions ranged from 1 to 12.7 km offshore, noting only those within 10 km were used for this study given this was the defined study area. The average singer distance from the visual observation station was between 5 and 6 km in all years. With the increase in density of migrating groups, there was some evidence that the migration corridor was more spread out. Group observations ranged from approximately 1 to approximately 20 km from shore in all years. The mean distance from shore was 7.4 and 6.6 km in 2003 and 2008 respectively, and 8.6 km in 2015.

Within each timeframe, the number of likely migrating males per day was a strong predictor of the number of males that sang per day (Fig. 3) in that more males per day equated to more singing males per day. On low density days (less than 10 males per day and less than 12 groups spread throughout the day), between zero and two singing males were recorded in the area. Singing males started singing alone and remained in the same area for approximately one to three hours. Once this singer had left the area, another singer would sometimes start later in the day. On high density days (more than 10 males per day and up to 90 groups migrating through the area), more than two singers were often recorded moving through the study area. On these days, multiple singers may have been in the area at any one time. They were usually separated by at least 2 km, and started and

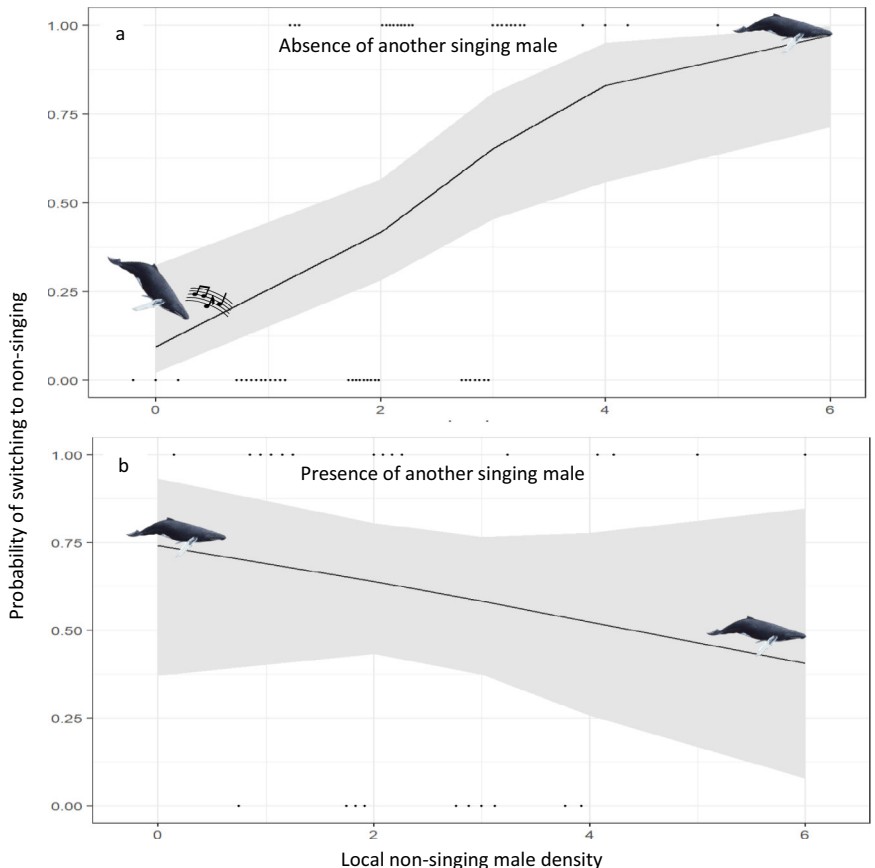

**Fig. 2 The probability of a lone focal whale switching from singing (0) to non-singing (1) as a function of local non-singing male density and behaviour.**
For display purposes results are separated into focal whales (*N* = 40) that switched in the absence (**a**) and presence (**b**) of another singing male. The line is the model estimate of the relationship (generalized linear mixed model assuming a binomial distribution) between the local non-singing density (*P* = 0.004), measured as the number of other presumed males within a 5 km radius of the focal male, and the presence of another singer (*P* = 0.003) including and the 95% confidence interval (grey area) on the probability a male humpback whale will switch from singing to non-singing behaviour. Raw datapoints are also included at the bottom (focal whales switched to singing) and top (focal whales switched to non-singing) of each graph. Icons illustrate singing and non-singing behaviour.

stopped singing at different times. Each singer would behave according to the proximate effect of local male density, where it would start and stop singing according to the number of other males within its social circle.

**Evidence of a shift in male mating tactics over time.** From the visual observations it was clear that the behaviour of the migrating groups visibly changed over time. In 1997, the number of observed non-singing (likely) males ranged from 1 to 17 per day and the number of observed singing whales ranged from 0 to 3 per day. During the 1997 study, groups with more than three adults (competitive groups), and the additional joining of whales to migrating groups, were rarely observed. By 2015, the number observed non-singing (likely) males ranged from 12 to 90 per day and the number of observed singers ranged from 0 to 6. Groups with more than three adults (competitive groups) were observed on most days, as were additional joins of males to groups. In other words, as the population increased, the number of males increased, and this led to increasing male-male physical competition as indicated by the increasing presence of multi-male competitive groups.

Up to six singing whales were recorded in a day in 2014/2015, compared to a maximum of three in 1997. However, when controlling for the proximate effect of male density, there was a significant decrease in the ratio of singing to non-singing whales in 2014/2015 compared to 2003/2004 (Fig. 4). For example, in

2003/2004, 10 migrating males equated to two singers. By 2014/2015, this proportion was reduced by half. This suggests a reduction in the number of migrating males that chose the singing strategy as the population increased post-whaling and a shift in the predominant male mating tactic towards physical competition.

**The effect of time on mating tactic payoff.** Each tactic likely has costs and benefits. Here, the tactic was assumed to be beneficial if any observed non-singing or singing male joined a group containing a female. This does not assume a successful mating took place, but that the male had a greater chance of mating given it was escorting, and therefore had access to, a female. Therefore, joining with a female acted as a proxy measure for breeding success. A female that was joined by a male was sometimes additionally joined by another male as described in the general observations. This was assumed to be a cost, given the group contained more males competing for the one female and the chances of any male being successful mating would be reduced. There is also an additional cost of increased likelihood of injury from physical competition.

The payoff for the non-singing and singing tactic was estimated each day by counting the number of sighted non-singing and singing joins, the number of sighted additional joins to the group, and dividing by the number of likely males. Scores ranged from 0 to 0.5 for each tactic. For example, a score of 0 for

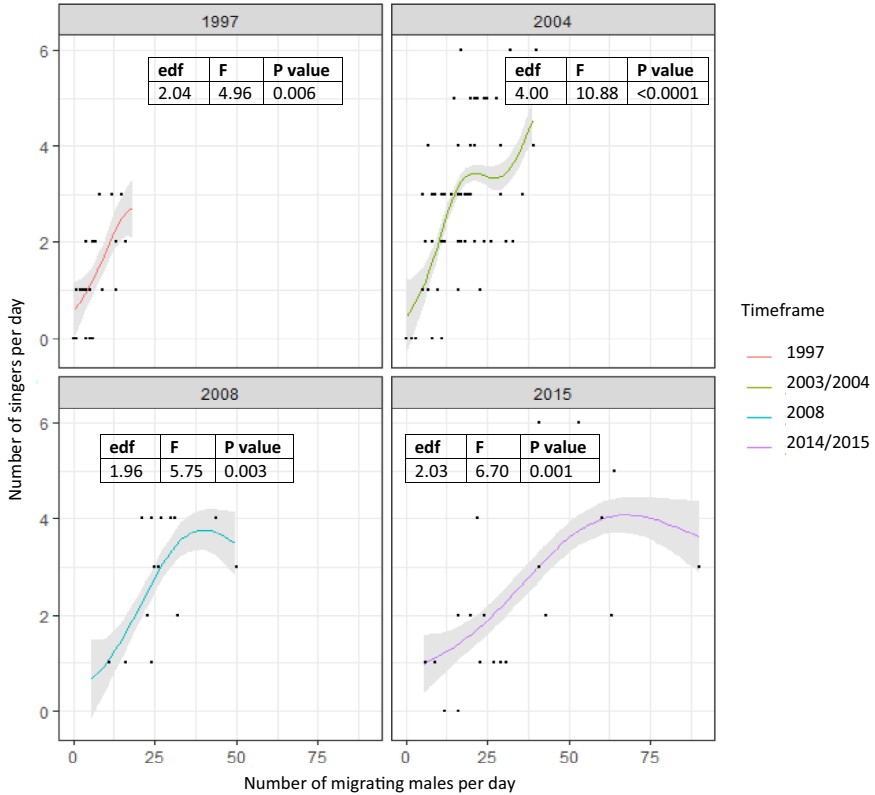

**Fig. 3 The changing relationship between the number of migrating males per day and the number of singing males per day over time.** Modelled estimate (using a general additive model structure including the number of migrating males and timeframe as the smooth term) of the relationship between the number of migrating males per day and the number of singing males per day within 1997 ($N = 28$ days; $P = 0.006$), 2003/2004 (categorised as 2004, $N = 61$ days; $P < 0.0001$), 2008 ($N = 14$ days; $P = 0.003$) and 2014/2015 (categorised as 2015, $N = 20$ days; $P = 0.001$). The line is the estimated smooth term for each timeframe along with the 95% confidence interval of the smooth plotted for the range of migrating males per day for each timeframe. Datapoints are the raw data. The significance the relationship of the number of singing whales to non-singing whales (shown as F and $P$-values) including the degrees of freedom (edf) reflecting the non-linearity of the curve.

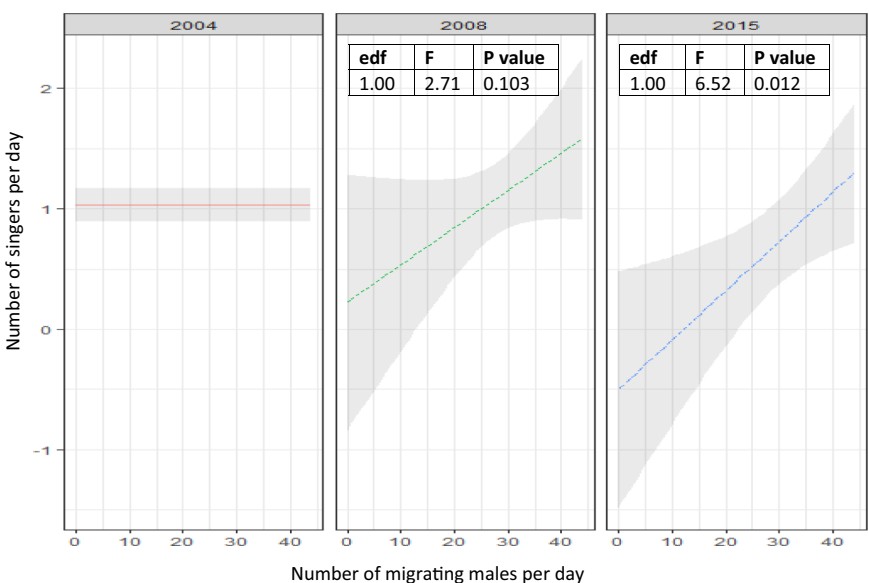

**Fig. 4 The difference in relationship between the number of migrating males per day and the number of singing males per day between 2003/2004 and subsequent years.** Modelled estimated difference in the number singing males per day per number of migrating males (including 95% confidence intervals) for 2008 ($N = 14$) and 2014/2015 (categorised as 2015; $N = 20$) compared to 2003/2004 (categorised as 2004; $N = 61$). The relationship between the number of singers and number of migrating males was held constant (at 1) in 2003/2004, allowing comparisons to be made in 2008 ($P = 0.103$) and 2014/2015 ($P = 0.012$). Differences were estimated using a general additive model, with timeframe as the ordinal effect and number of migrating males as the covariate (presented as F and $P$-values with edf, reflecting the non-linearity of the curve).

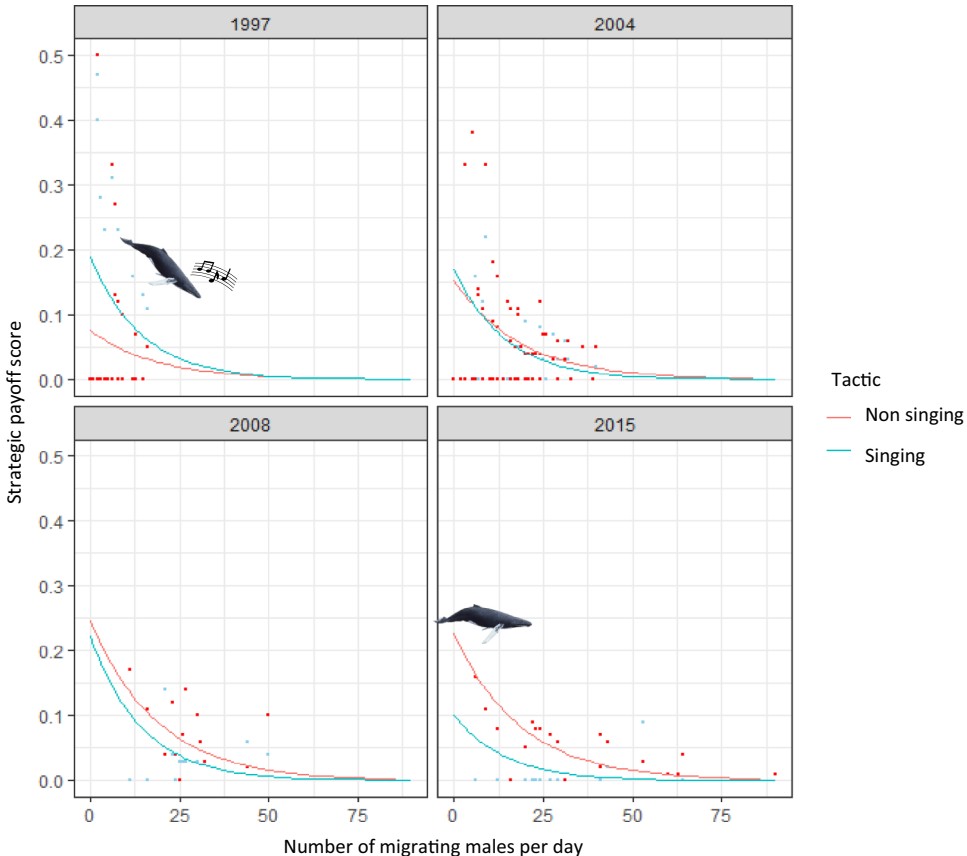

**Fig. 5 The changing relationship between the payoff for each tactic (singing and non-singing) and the number of migrating males over time.** Modelled relationship (assuming a zero-inflated negative binomial model structure) between the tactical payoff score (number of joins minus the number of additional joins for non-singing and singing whales) for non-singing (red) and singing (blue) males and the number of migrating males per day (line) in 1997 ($N = 28$), 2003/2004 (categorised as 2004, $N = 61$), 2008 ($N = 14$) and 2014/2015 (categorised as 2015, $N = 20$). Raw data are included as datapoints. Singing payoff scores did not significantly differ between timeframes, non-singing payoff scores significantly increased between 1997 and 2003/2004 ($P = 0.0133$), 2008 ($P = 0.0037$) and 2014/2015 ($P = 0.0005$). Icons represent the greater payoff of the singing tactic in 1997, and the greater payoff of the non-singing tactic by 2014/2015.

the singing tactic meant no singers were sighted joining during the day. A score of 0.5 for the singing tactic meant that, out of (for example) two migrating males, one sang and joined a female with no additional join (cost) to this group. If zero non-singing joins were observed that day, then the singer tactic outweighed the non-singer tactic.

We found that the estimated payoff for both the singing and non-singing tactic decreased as the number of daily migrating males per day increased, likely due to increased competition for available females (Fig. 5). Over time, as the population increased, the relative payoff of each tactic within timeframe changed. In 1997, the singer tactic had a significantly (Generalized Linear Model (GLM); $N = 28$; $Z = 2.859$, $P = 0.004$) greater payoff than the non-singing tactic. This is because, per day, a singing male was 1.8 times more likely to be sighted joining another group compared a non-singing male. Additional joins to these groups were rarely sighted meaning there was little cost to this tactic. In 2003/2004 there was no significant difference in payoff between the two tactics (GLM; $N = 61$; $Z = -0.851$, $P = 0.395$). By 2008, non-singer joins were 2.3 times more likely to be sighted than singer joins, and by 2014/2015, non-singing joins were 4.8 times more likely to be sighted than singer joins. Even though additional joins to these non-singing groups were also more commonly sighted in later years, the number of joins (benefit) outweighed the number of additional joins (cost), meaning a significantly higher payoff for this tactic in later years (GLM;

$N = 14$; $Z = -2.430$, $P = 0.0151$ for 2008 and $N = 20$; $Z = -3.763$, $P = 0.0002$ for 2014/2015).

When comparing singer payoff between years, we found that it did not significantly change (Fig. 5; Table 1). In other words, within each timeframe, the relationship between the number of migrating males per day, and the payoff for the singing tactic, was the same despite the increase in population. The non-singer payoff, however, significantly increased over time (Fig. 5; Table 1). This difference in payoff was especially obvious on days when there were a relatively low number of migrating males. On days in which there were less than 15 migrating males per day, the payoff for non-singing males in 1997 was likely to be zero. By 2014/2015, with a similar number of migrating males, the payoff for a non-singing male was never zero given at least one non-singing join was sighted. Note, this does not necessarily translate to increased breeding success, but does mean males had an increased chance of accessing a female.

## Discussion

Various studies have shown that behavioural plasticity in mating strategies can increase their ability to cope with anthropogenic impacts[35]. While rapid environmental change can be a strong selective force for individual-level behavioural plasticity, a fast population-level response to this change can be beneficial in lowering its extinction risk[36,37]. However, assessing how animals

**Table 1 Tactical payoff score for a singing and non-singing whale as a function of male density over time.**

| Strategy | Parameter | Estimate | Std. Error | z value | p-value |
|---|---|---|---|---|---|
| Singing | Intercept | 3.175 | 0.114 | | |
| | Migrating males | −0.053 | 0.006 | −9.002 | <0.0001 |
| | Intercept (1997) | −0.436 | 0.387 | | |
| | 2003/2004 | 0.256 | 0.466 | 0.549 | 0.583 |
| | 2008 | −0.503 | 0.715 | −0.704 | 0.481 |
| | 2014/2015 | 1.208 | 0.631 | 1.915 | 0.056 |
| Non-singing | Intercept | 3.074 | 0.116 | | |
| | Migrating males | −0.043 | 0.005 | −8.401 | <0.0001 |
| | Intercept (1997) | 0.916 | 0.418 | | |
| | 2003/2004 | −1.224 | 0.493 | −2.484 | 0.0130 |
| | 2008 | −3.607 | 1.242 | −2.904 | 0.0037 |
| | 2014/2015 | −2.700 | 0.776 | −3.481 | 0.0005 |

Results of the zero-inflated negative binomial model comparing the tactical payoff score (benefit of joining a presumed female minus the cost of being additionally joined by another male) for a singing and non-singing whale over time including the number of migrating males as a covariate. The table indicates the estimated difference in score between 1997 and subsequent years, the standard error of the estimate and the significance of the difference in scores (as z and p values).

respond to unprecedented changes is difficult to do in wild animal populations given the logistical challenges of collecting the necessary data. Here, using an 18-year dataset, we have evidence that increasing local male density, and a severe change in population density, influenced the frequency, and likely fitness pay-off, of alternative mating tactics in a wild animal, the humpback whale.

An animal's ability to modify its mating behaviour in response to changes in its immediate social environment relies on its capacity to assess mate and competitor density[38]. The animal will maximise its reproductive potential if it chooses the best tactic according to the balance between the number of available mates, and the competition for those mates. Mating tactics of many species, including fish[11] and amphibians[12] are expressed according to the density of competing males. Considering this, we first tested the influence of local male density on the likelihood of observing male humpback whales singing or switching to the non-singing tactic. Our results demonstrate that male humpback whales follow suite, in that males choose to sing in low male competitor densities and switch to non-singing (which includes physical competition) behaviour when local male density increases. Earlier studies in this population found that males were less likely to sing whilst escorting a female in the presence of other males due to the increased risk of attracting competition for that female[32,34]. In other words, males are likely maximising their reproductive potential by singing when the risk of attracting male competition is low. Switching to non-singing behaviour may reduce the risk of attracting competition, given song is a loud acoustic signal. To support this idea, the current study, as well as previous studies in this study site[30,32,34], documented singing whales being joined by another male. These singing whales started to sing in low male densities, then stopped singing as soon as they were joined. This behaviour has also been documented in Hawaiian breeding grounds[29,31], where song is thought to function as a method of dominance sorting and/or a means of cooperation during mating. Whether being joined by another male is a cost (e.g., male-male competition) or benefit (e.g., male-male cooperation) to singing behaviour is difficult to ascertain with such a small sample size. Of note is that in this study, all observed male-male pairs quickly split following the join, meaning, in these cases, it is unlikely to be indicative of male cooperation. We can, however, assume there is a benefit to the joining male, given joining with the singer seems to be directed 'intentional' behaviour. The benefit to the joining male may be to cause the singing whale to stop and to move out of the area, thus increasing the joiner's chances of associating with available

females. However, regardless of the cause and effect of male competition on their singing behaviour, local male density was found to be a significant proximate factor in determining whether humpback males sang or ceased to sing. Their mating strategy is therefore behaviourally plastic and likely to be conditional on male competition.

Large changes in population density can alter the relative fitness costs and benefits of mating behaviours, given they are sexually selected. This can affect female fecundity, reproduction rates, offspring fitness[39], and result in the collapse of the population. For example, the poaching of male saiga antelope (*Saiga tatarica*) horns resulted in a female-biased sex ratio, which led changes in their sexual selection behaviour and population collapse[40]. However, density-related population collapse is not always the case, and the adjustment of mating behaviours to population density may prevent a collapse. Some species, such as rodents[41] and red grouse (*Lagopus lagopus scoticus*[42,43]) exhibit density-dependent changes in the strength of selection towards aggression. At low densities, the fitness benefits of locating a female outweigh the benefits of using an aggressive competitive tactic. As the population density grows, the increasing cost of aggression (e.g., injury) is outweighed by the benefits of increased access to females[44]. In this study, as the humpback whale population increased post-whaling, we documented a population-level "switch-point" in the abundance and success rate of the non-singing over the singing mating strategy. This suggests males modified their mating tactics in response to the changing selective pressure of population density. During the early post-whaling years, the low population density would have made finding and competing for a suitable mate challenging. We showed that, under such circumstances, singing was the most common and successful strategy in terms of joining with a group containing a female. As the population steadily recovered, and competitive groups consisting of multiple males competing for a female were sighted more often, we found that the non-singing (physical) tactic became the more dominant tactic. This status-dependent "switch-point" is expressed in many species[45] and is indicative of changes in the relative fitness of divergent tactics as environmental and demographic parameters change.

It is also important to acknowledge that the age composition of the recovering humpback whale population would have changed together with the observed increased in population size. This is because whaling likely targeted, and wiped out, mature animals. The change in population dynamics in terms of age structure may be particularly important if older males are more likely to choose the aggressive tactic, as found in other species[5]. If we assume

whaling wiped out mature males, then it would have taken some time for older males to reappear within the population dynamics. This makes it difficult to identify the influence of age on humpback mating tactics. Therefore, it would be interesting to carry out this study once more to determine if the benefits to aggression reach a plateau, in that costs of aggression continue to increase, while access to breeding females decrease[46]. If that was the case, perhaps males will once again adopt the passive singing strategy, or invest in alternative traits, such as swim speed, body size, or weaponry, that increases their fitness[47].

In summary, despite being whaled to almost extinction, this species has made an unprecedented recovery[20] meaning humpback whales must have successfully adapted to the extreme post-whaling changes in their population structure. Our results found that male humpback whales are likely to have a density-dependent behaviourally flexible mating strategy which is comprised of two tactics: singing behaviour, and non-singing, which can lead to aggressive, behaviour. In this population, at a proximate level, males switched between singing and non-singing behaviour according to the local density of other males. At a population level, over time, males were more likely to use the non-singing strategy. Animals which can adapt in this way are thought to be better suited a changing social environment[48,49]. Therefore, the fact that there is likely to be a density-dependent phenotypic plasticity in humpback male mating tactics in this population may have lowered their extinction risk and contributed towards their recovery.

## Methods

**Study area and general observations.** Four datasets, equating to four post-whaling timeframes, were used for this study: 1997 (32 years post-whaling), 2003/2004 (38/39 years post-whaling), 2008 (43 years post-whaling) and 2014/2015 (49/50 years post-whaling). Data collection for each timeframe occurred during the annual migration of humpback whales, from breeding grounds in the Great Barrier Reef, to feeding grounds in the Antarctic Ocean. The study site was located off the coast of Peregian Beach (north of Brisbane, in Queensland, Australia), which was approximately one-third of the way along their return migration route. Here, humpback whales were still exhibiting breeding behaviours, such as singing, males joining females as escorts, and males forming competitive groups around a central female. Field work took place in September and October of each year. Generally, the number of migrating groups increased per day to peak during late September and early October. Numbers then gradually fell until the end of the migration.

For this study, a *group* was defined as cluster of whales within approximately 100 m of each other that were diving and surfacing together (as estimated by the land-based visual observers). Groups were constantly changing membership with animals joining and splitting from the group and tend to move at different speeds, and in different directions, whilst making general progress southwards. Groups, unless joining together, were separated by at least 2 km, meaning it was relatively easy to keep a separate track of each group (see below).

Acoustic recordings were made from three to five hydrophone buoys moored in 18–28 m of water and arranged in a line or T-shaped array (Fig. 6). Each hydrophone buoy consisted of a surface buoy containing a custom-built pre-amplifier (+20 dB gain) and 41B sonobuoy VHF radio transmitter. A High Tech HTI-96-MIN hydrophone with built-in +40 dB pre-amplifier was suspended approximately 1 m above each buoy's mooring. Signals were received onshore at a base station 1.5 to 2.5 km away using a directional Yagi antenna and type 8101, four-channel sonobuoy receiver. Singing whales were located by cross-correlating the same song sound arriving at the different hydrophones to determine time-of-arrival differences. These differences, together with an accurate knowledge of the positions of the hydrophones, were then used to determine the most likely location of the singer. Singers generally move slowly and calculating an acoustic position approximately every 10 min produced a detailed track of the singer.

Migrating groups were tracked visually (7am to 5pm, weather permitting) from a land-based elevated survey point, Emu Mountain (73 m elevation). A theodolite (Leica TM 1100) was used in conjunction with a notebook computer running *Cyclopes* software (E. Kniest, Univ. Newcastle, Australia) to track the groups in real-time and note group behaviours. The field of view was approximately 20 km in a north/south direction and 10 km offshore (Fig. 6). Humpback whale groups were observed *ad libitum* and tracked by teams of five people. When whale groups surfaced, the observers called the sighted behaviour, compass bearing, and angle from the group to the horizon (in reticules). Each observation included group identification letter, the time, group size and composition, whether a calf was present, direction of travel, and group location, either by using a binocular reticular measurement or a theodolite measurement. Joining and splitting animals were also

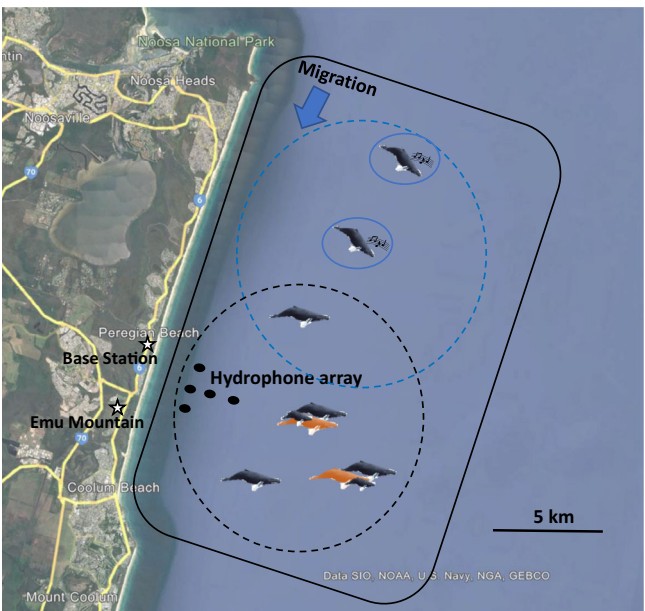

**Fig. 6 Outline of the study site including the range of visual observations and the position of the acoustic tracking array.** Illustrating the study site at Peregian Beach, north of Brisbane, east coast of Australia. The map indicates the position of the land-based station (Emu Mountain) and the acoustic base station along with the position of the 5-buoy hydrophone array. The outline designates the study area. Whales moved in a southerly direction through the area daily. Whale icons illustrate acoustically tracked singing whales (circled in blue) and visually tracked presumed males (black), females (orange), and calves (small black). The 5 km social circle radius for a focal singing (blue circle) and a non-singing (black circle) male are also illustrated. The map is taken from "Google Earth" with permission to print without the need to submit a request (Brand Resource Center | Products and Services - Geo Guidelines (about.google)).

noted. A *join* was defined as one of more animals actively moving towards a group to surface within 100 m and then match the group surfacing times. Examples of this include an individual singing or non-singing whale actively moving towards, and then joining, another individual or group of whales. If more animals subsequently moved in and joined the group, this was termed an *additional join* to that group. These additionally joined group usually comprised of a female-calf and more than one male escort, or three or more adults, with additional joiners highly likely to be male ([21,25,26], supplementary results). On rare occasions a singing whale remained in one place but was joined by another individual. This was termed an *additional join* given there was no evidence the singer actively moved to join this animal. However, the rarity of these occurrences meant the allocation of this behaviour to *additional join*, rather than *join*, had no influence on the results.

Some of the migrating animals were biopsied during the day for post-field later sexing. Note biopsied animals were sometimes part of different studies occurring at the field site[30,50] and were not necessarily the animals used in this study. However, these biopsy results were used to test assumptions made in this study regarding the sex of joining whales and whales within the observed groups (supplementary results and supplementary note). Weather was noted hourly.

### Statistics and reproducibility

*Defining the proximate effect of male density on individual mating tactics.* For this analysis, a specific period, the 2003/2004 dataset, was chosen as it had the most instances of identified singers and non-singers. Within this timeframe, whales were migrating through the study area at sufficiently low density to avoid confusion. After 2004, it became increasingly difficult to focally follow males.

First, for singing males ($n = 86$), their location within the study area was recorded at the start of singing using the acoustic array. Whilst singing they remained in the same location or meandered slowly within a small area. Non-singing animals that were observed to join a group ($n = 31$) were assumed to be male ([21,25,26,30], supplementary methods and supplementary results). For these joining animals, visual observations were backtracked for 10 to 15 min until they were sighted alone. They were only included in the analysis if they could be definitively backtracked using visual (theodolite) observations, with no opportunity for confusion with other whales in area (i.e., no other whales within 2 km).

For each unaccompanied focal male, the number of, and roles, of other presumed males within 5 km radius from the focal whale (Fig. 6) was used as a measure of local male density. The 5 km radius was termed *social circle* and was chosen as the most likely communication space for their acoustic signals[51]. For singing focal whales, their social circle was estimated using their location when they began to sing. For non-singing focal males, their social circle was estimated using the backtracked theodolite position to when it was first sighted alone. Next, all groups within the 5 km social circle of the focal whale, along with each group composition (singing animal, lone animal, female and calf pair, female-calf and escort number, adult-only group with the number of adults) were recorded at that timepoint. It was not logistically possible to biopsy and sex all migrating animals, therefore, to estimate the number of males within their social circle several assumptions were made. These assumptions were also tested using a biopsy study carried out in the area (supplementary methods and supplementary results). Female-calf pairs were discounted as it was assumed all adults with a calf were female. It was assumed that female-calf pairs were being escorted by males ([21,25,26], supplementary methods and supplementary results). Groups of multiple adults were assumed to be comprised of a likely single male, principal male escort and secondary male escorts or challengers ([21,25,26], supplementary methods and supplementary results). Lone animals not involved in any group interactions, and not singing, were given a 70% chance of being male (supplementary note). Animals within adult pairs were given a 70% chance of being male given the likelihood of having a mix of female-male pairs and male-male pairs ([21,30], supplementary results and supplementary note).

All analysis models were carried out in R (version 3.4.0). The first analysis aimed to determine if the likelihood of first observing the focal individual as a singing or non-singing male was significantly related to local male density, as determined by the number of males within a 5 km radius, termed social circle. Singing whales were allocated a 0 and non-singing whales were allocated a 1. A generalised linear model structure was used, assuming a binomial distribution. Likely males within their social circle were divided into non-singing and singing males (to delineate tactics) and these were included as the two covariates.

$$\text{Singing}(0) \text{ or Non-singing}(1) \sim \text{Non-singing males 5 km} + \text{Singing whales 5 km}$$

Each focal male was an independent sample given males were migrating southwards and extremely unlikely to back-track into the study area and therefore be resampled. Significance was set at $p < 0.05$. Effects were plotted as estimates from the model fit along with the 95% confidence intervals.

The second analysis used focally followed individual males that were observed to switch between tactics. This required animals that could be observed and focally followed, without risk of confusing them with other whales, and that switched tactic during this observation period. Of the 117 focal males, 40 met these criteria. Their social circle was quantified when they were first observed along and singing and again as soon as they stopped singing. To test if focal males were more likely to switch tactic in increasing local male density (the number of non-singing and singing males in the area), a generalised mixed model structure was used, assuming a binomial distribution, and including focal male ID as the random effect to accounted for repeated measures within animals.

$$\text{Singing}(0) \text{ or Non-singing}(1) \sim \text{Non-singing males 5 km} * \text{Singing whales 5 km} + (1|\text{ID})$$

Significance was set at $p < 0.05$. Effects were plotted as estimates from the model fit along with the 95% confidence intervals.

*Frequency of alternative mating tactics.* Given the large increase in the humpback whale population post-whaling, the third analysis aimed to determine if the ratio of the alternative mating tactics changed at the population level. Here, daily observations, rather than individual records, were summarised. Each day ($N = 123$) comprised of 10 h of combined land-based and acoustic observations. For each day, the number of singing whales was counted using the acoustic recordings and concurrent sightings of the singing whales (Fig. 6). Every migrating group was visually tracked as it moved through the area and allocated a group composition as described above. Group compositions were then used to estimate the number of migrating whales per day that were likely to be male, as detailed in the supplementary results and supplementary note. This gave, for each day, a total number of singing whales and an estimated total number of migrating males.

The number of singers per day (response variable) was then correlated against the number of males migrating through the area in a day within each timeframe (categorised as 1997, 2003/2004, 2008, 2014/2015). As these were count data, ranging between 0 and 6 singers, with evidence of underdispersion, a quasi-Poisson distribution was assumed, where the $p$-values and confidence intervals were adjusted using an estimated dispersion parameter. A generalised additive model (gam) structure was used given it does not assume a fixed relationship between the response variable and covariates (mgcv package[52]). The two covariates were timeframe and the number of migrating males within each timeframe. The latter was the smooth term modelled separately for each timeframe. Significance was set at $p < 0.05$.

$$\text{No. singers} \sim s(\text{Migrating males by Timeframe})$$

*Evidence of a shift in male mating tactics over time.* To determine if the mating tactics of males shifted post-whaling, we compared the ratio of singing to migrating whales across timeframes. We hypothesised that if mating tactics did not change over time, the ratio of singing to migrating males (i.e., the proximate effect of local male density) would be stable. If males were shifting their tactics towards singing or physical competition, the ratio would change. Note, there was no evidence that the proportion of migrating adults that were likely to be male changed over time (supplementary note).

A gam structure was used (mgvc package[52]).The 1997 dataset was not included given the low numbers of migrating whales per day which were not comparable with later years. In addition, given there were no days in which more than 50 non-singing whales migrated through the area in 2003/2004 or 2008, the number of migrating males was capped at 50. Year was included as an ordered variable, meaning the model output was the difference between the smooth estimated for the reference level (2003/2004) and those from subsequent years. Simply put, the relationship between the number of migrating males and singing whales in 2003/2004 was held constant, and the difference in the number of singing whales for each number of non-singing whales was then estimated in subsequent years. Significance was set at $p < 0.05$.

$$\text{No. singers} \sim \text{ordinal Timeframe} + s(\text{Migrating males by ordinal Timeframe})$$

*The effect of time on mating tactic payoff.* A payoff score for an individual non-singing, and singing, male was created per day based on the number of observed non-singing joins (likely benefit) and additional joins (likely cost), number observed singing joins (likely benefit) and additional joins (likely cost), corrected for the number of migrating males to give an adjusted score per individual. This relied on general visual observation data, meaning males did not have to be focally followed (which would not have been possible). However, because singing and non-singing joins and additional joins could be identified (using acoustic tracking data to identify which of the tracked animals was singing), individual payoff scores could be separately estimated for both tactics.

These individual non-singing (ns) and singing (s) payoff scores were estimated as:

$$(\text{Total ns joins} - \text{Total ns additional joins})/N$$

$$(\text{Total s joins} - \text{Total s additional joins})/N$$

where $N$ was the total number of migrating (presumed) males per day. This gave an estimated payoff score for a singing male and non-singing male for each day. These payoff scores were then compared within and between each timeframe. To do this a series of generalised linear models were ran.

First, to test if one tactic was more successful than the other within each timeframe, the following model was used where a zero-inflated negative binomial distribution was assumed due to overdispersion and an excess of zeros (glmmTMB package[53]):

$$\text{Payoff Score} \sim \text{Tactic} + (1|\text{Day})$$

Payoff score was then estimated as an individual score each tactic, to test is one was consistently greater than the other, and the random effect of day was included to account for the repeated measures (i.e., the payoff score for each tactic was estimated for each day resulting in a paired comparison). A separate model was run for each timeframe (i.e., for the 1997, 2003/2004, 2008 and 2014/2015 datasets).

Then, estimated daily payoff scores were then compared between timeframes to determine if either tactic become more, or less, successful over time. Given local male density also increased over time, the number of migrating males per day was included as a covariate. The following models were used assuming a zero-inflated negative binomial model structure (pscl package[54]) given this was count data that exhibited overdispersion, and an excess of zeros:

$$\text{Singer Payoff Score} \sim \text{Migrating Males}|\text{Timeframe}$$

$$\text{Non-singer Payoff Score} \sim \text{Migrating Males}|\text{Timeframe}$$

Singer and non-singer payoff scores were modelled separately. Model results are plotted as the estimated relationship between the payoff score per individual for a singing and non-singing male and the number of migrating males. These were plotted separately for each timeframe.

**Reporting summary**. Further information on research design is available in the Nature Portfolio Reporting Summary linked to this article.

## Data availability

Datasets are stored on the University of Queensland's Research Data Management system and UQ eSpace (https://rdm.uq.edu.au/files/fe8174b0-6c4a-11ed-bf3b-3b320cd4f16b;[55]). Access can be requested via the UQ eSpace website by following the links provided.

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

## Acknowledgements

This work would not have been possible without Professor Michael Noad (School of Veterinary Science, University of Queensland) who began work on singer behaviour at this study site in 1995. Professor Noad also developed the real-time acoustic tracking equipment. The authors would also like to acknowledge and thank Dr Eric Kniest (formerly of the University of Newcastle), who developed the tracking programme, *Cyclopes*. Finally, the authors would like to thank the numerous volunteers that donated their time and energy to the multiple projects which were the source of these data; specifically, the Humpback Acoustic Research Collaboration (HARC) project (2002 – 2008, supported by the U.S. Office of Naval Research, the Australian Defence Science and Technology Organisation and the Australian Marine Mammal Centre) and the Behavioural Response of Australian Humpbacks to Seismic Surveys (BRAHSS) project (2010 to 2015, supported by the Joint Industry Program E&P Sound and Marine Life).

## Author contributions

RD conceived the study design, collected, and analysed the data, and authored the paper. CF reviewed and edited the paper and provided intellectual input into the conclusions and broader significance of the results.

## Competing interests

The authors declare no competing interests.
