## [Peer Review File · Communications Biology]

Reviewers' comments:

Reviewer #1 (Remarks to the Author):

Thank you for the opportunity to read and contribute with this important constructive process in science. This paper constitutes in a very good contribution to the behavioral ecology field. It claims that due to extensive whaling, male humpback whales from the eastern Australia population have shifted their mating tactic system, basically from singing to physical competition. For this, using generalized linear modeling, authors analyzed a dataset of 18 years from a land-based and acoustic monitoring to reach the conclusion of changing mating tactics for that particular population. The current manuscript is well designed and results are innovative. Attached, it follows a minor editing list, with some highlights that could improve even more the general article.

Reviewer #2 (Remarks to the Author):

Overall, I think this is an outstanding article. This sort of long-term examination of humpback whale mating strategies is as needed as it is scarce. I commend the authors for the effort that so evidently supported the monitoring program for over 18 years. The methodology is adequate and the statistical analyses sufficient to answer the research questions. Most of the content within the sections flows very well, making this quite an enjoyable read. My main concern would be the assumptions regarding payoffs of each tactic, as I find them too speculative. The supplemental material was quite helpful in dissipating my concerns regarding sexing predictions. While I assume limiting to just two strategies (competition within mating groups and singing) better fits the data available, leaving the escorting of maternal females with little to no mention is inadequate in my opinion. There are a few important suggestions when it comes to structure, but the vast majority of my comments are related to format, grammar and clarity, and are ultimately to the discretion of the authors and the editing team.

Reviewer #3 (Remarks to the Author):

The paper presents research from an extraordinary dataset. I don't think there would be a dataset comparable to this in terms of monitoring the recovery of a whale population. The idea behind the study is fantastic and ultimately this study should be published. However, there needs to be some clarity around some sections as they were confusing with the message that was coming across. Abstract could be restructured by bringing the sentence starting line 19 forward in the abstract, as this is pertinent information to the results that is presented before it. Also, +20000 should be >20000. The first section of the results is where I struggled the most and needs to be re-written to be more concisely to improve the clarity. The last paragraph of this section provides a concise summary, which is the main thrust of this section, but that clarity is lost within the text of how it is currently written.

Line 232 – “another singer may have started.” What is meant by “may?” This is an inaccurate term the way it is used. If a singer left the area and you can't prove another singer starts then delete this sentence.

Line 285 – Sentence starting “By 2014/2015...”. The graph suggests that 10 males per day equates to 0 singers not one singer.

Overall, I am struggling with the “in text” interpretation (lines 284-285) of figure 4. 2003/2004 10 males equated to two singers' figure 4 suggests 1 singer, not 2. Also, y-axis label on figure 4 is missing “day.”

Figure 6 legend – females (black)? Should this be females (red).

Reviewers' comments:

Reviewer #1 (Remarks to the Author):

Thank you for the opportunity to read and contribute with this important constructive process in science. This paper constitutes in a very good contribution to the behavioral ecology field. It claims that due to extensive whaling, male humpback whales from the eastern Australia population have shifted their mating tactic system, basically from singing to physical competition. For this, using generalized linear modeling, authors analyzed a dataset of 18 years from a land-based and acoustic monitoring to reach the conclusion of changing mating tactics for that particular population. The current manuscript is well designed and results are innovative. Attached, it follows a minor editing list, with some highlights that could improve even more the general article.

Response: the authors thank the reviewer for this positive response and also for taking the time to submit much appreciated edits. The following addresses each suggestion:

REVIEW

Pag 1 – LINES 23-25

“This individual-level plasticity in male humpback whale mating tactics likely minimised their risk of extinction following a dramatic change in their social landscape due to whaling”. (THIS IS BROAD IDEA TO BE AFFIRMED WITH THE PRESENT RESULTS. FIRST, WE DON’T KNOW HOW THE MATING SYSTEM WAS BEFORE THE EXTENSIVE WHALING OR HOW “PLASTICITY” IT COULD BE IN A TEMPORAL SCALE. SECOND THAT OTHER INFLUENCES, THAN MALE DENSITY COULD BE INFLUENCING THE MATING SEASON AND THIS WOULDN’T BE THE ONLY CONDITION TO MINIMISED THEIR RISK OF EXTINCTION. ONE KEY POINT I THINK IS TO CONSIDER THE IMPORTANCE OF THE RESULTS FOR A SPECIES LEVEL, BUT ALSO THAT IT WAS OBSERVED IN THIS PARTICULAR POPULATION, AS THE HUMPBACK WHALE IS A COSMOPOLITAN SPECIES PRESENTING A LARGE REPERTOIRE OF PLASTIC BEHAVIOUR AND STRATEGIES ELSEWHERE.

Response: point taken but argue that it must have been a contributing factor since there is no argument that their social landscape dramatically changed post-whaling due to the decimation of numbers and they have plasticity in their mating tactics. Softened the statement by adding ‘contributed to’.

As for the second point, this was addressed in one of the final review responses for this reviewer.

PAG 3 – LINE 52 – “The cessation of whaling has resulted in the recovery of many populations. One example is the eastern Australian population of humpback whales (*Megaptera novaeangliae*).’ (IT WOULD BE APPROPRIATED TO MENTION ABOUT A SPECIES LEVEL BEFOREHAND A POPULATION LEVEL. HERE IT IS SUGGESTED THAT AUTHORS SHOULD INCLUDE A PREVIOUS MENTION ABOUT THE STATUS OF THE SPECIES WORLDWIDE (HUMPBACK WHALE) AND THEN MENTION A SPECIFIC POPULATION

Response: done

PAG 3 – LINE 54 – A whale population has been WHALED, sounds awkward...maybe

replace for HUNTED OR EXPLORED

Response: done

PAG 3 – LINE 57-“unprecedented recovery in that (WHICH) the population number has”...

Response: done

PAG 3 – LINE 59-61

“Assuming humpback whales live 60 for approximately 60 years, ARE ONLY NOW (2020s) is the population likely to include animals 61 that have lived their entire lifespan.” SOUNDS AWKWARD” MAYBE EXCLUDE THE WORD ARE? OR REPLACE FOR AND?

Response: done

LINE 68 - to attempt to breed with her (ATTEMPTING)

Response: changed ‘to’ to and as this suggesting wasn’t grammatically correct

PAG 5 – LINE 111-112

Earlier studies carried out in this field site found that singing males were less likely the sing (TO SING?) in the presence of other competing males

Response: done

PAG 7 – LINES 158-160

The line is the model estimate of the relationship (generalized linear model assuming a binomial distribution, $P = 0.0002$) including and (DELETE AND) the 95% confidence interval (grey bar).

Response: done

PAG 8 – LINE 190

In other words, individual mating tactics (PLURAL) was (WERE) dependent on both the number,

Response: done

PAG 10 LINE 235-236

Singing males usually started singing alone and remained in the same area for a period (HOW LONG? PLEASE BE MORE PRECISE).

Response: done

I SUGGEST TO MAKE IT CLEAR IN THE MAIN TEXT THE QUESTION OF HOW TO COUNT MALES IN THE FIELD (COUNTED AS A SINGER OR AN APPROACHING ANIMAL TO A FEMALE-CALF PAIR, FOLLOWING YOUR DESCRIPTION IN THE SUPP. RESULTS.

Response: this was in the methods section as per the journal’s preference of having the results first followed by the methods. To put this in the main results text would mean repeating the

methods.

I FOUND THIS SUPP. RESULTS VERY INFORMATIVE. EVENTHOUGH AUTHORS COULD BETTER CLARIFY THE BEHAVIOURAL/GENETIC CLASSIFICATION FOR MALES IN THE MAIN TEXT.

Response: as the reviewer points out, this was in the supplementary results. The reason being, it was almost a study onto itself and therefore would require a lot of additional text in the main paper. The sex assumptions are outlined in the main text and if readers require further information on how valid those assumptions are, it is available for reading in the supplementary.

PAGE 19 LINES 422-423

“This behaviour 423 has also been documented in Hawiian (HAWAIIAN) breeding grounds”

Response: done

PAGE 21 LINES 459-460

“The European earwig (*Forficula Auricularia*), for instance, will transition between monomorphic and dimorphic populations based on changes in population density (Tomkins and Brown, 2004)”. I DON’T THINK COMPARING AN INSECT POPULATION TACTIC WOULD BE APPROPRIATED HERE, AS THEY ARE SO DISCTINCT AND PARTICULAR COMPARED TO THE WHALES...I’D SUGGEST RESTRICTING THE DISCUSSION ON VERTEBRATE STRATEGIES

Response: deleted

PAGE 21 LINES 469-470

“Mature males would have, however, only began to appear in our dataset in 2015 (last year of the study) making it difficult to identify the influence of age on humpback mating tactics”. HOW CAN AUTHORS AFFIRM THAT? WHAT PARAMETERS WERE ESTIMATED FOR MATURE/UNMATURE MALES? SINGING?? PLEASE PROVIDE FURTHER EXPLANATION ON THAT

Response: changed to “If we assume whaling wiped out mature males, then it would have taken some time for older males to reappear within the population dynamics. This makes it difficult to identify the influence of age on humpback mating tactics.”

PAGE 21 LINES 485-487

“Therefore, the fact that there is likely to be a density dependent phenotypic plasticity in humpback male mating tactics may have lowered their extinction risk and contributed towards their recovery”. PLEASE SEE COMMENT AT THE END OF THE ABSTRACT. THIS IS REALLY AN EXCITING FIDING BUT SHOULD BE REGARDED FIRST TO A POPULATION LEVEL (EAST AUSTRALIA POPULATION), AS THE SPECIES PRESENTS A GREAT PLASTICITY WORLDWIDE, AND DISTINCT LOCAL ENVIROMENTS COULD SHAPE THE MATING TACTICS DIFFERENTLY.

Response: done and changed.

PAGE 22, LINES 503-504

For this study, a *group* was defined as cluster of whales within 100 m of each other that were 504 diving and surfacing together. HOW WAS DISTANCE ESTIMATED? PLEASE

PROVIDE FURTHER INFORMATION

Response: estimated by the visual observers whilst viewing the group – added.

PAGE 23, LINE 514

Humpback whale groups were observed *ad lib* (*AD LIBITUM*)

Response: done

PAGE 25 FIGURE 6

Although illustrative and well design for the study details, the background map does not present a good definition and I think would not be appropriated to be published like this. Authors could please provide a better-quality background map to add the study particularities upon it.

Response: background map changed to a google earth map.

PAGE 25, LINES 575-576

Whale icons illustrate acoustically tracked singing whales (circled in blue) and 576 visually tracked presumed males (black), females (black) – (WOULD YOU MEAN LIGHT BROWN OR RED??), and calves (small black).

Response; yes and changed

Reviewer #2 (Remarks to the Author):

Overall, I think this is an outstanding article. This sort of long-term examination of humpback whale mating strategies is as needed as it is scarce. I commend the authors for the effort that so evidently supported the monitoring program for over 18 years. The methodology is adequate and the statistical analyses sufficient to answer the research questions. Most of the content within the sections flows very well, making this quite an enjoyable read. My main concern would be the assumptions regarding payoffs of each tactic, as I find them too speculative. The supplemental material was quite helpful in dissipating my concerns regarding sexing predictions. While I assume limiting to just two strategies (competition within mating groups and singing) better fits the data available, leaving the escorting of maternal females with little to no mention is inadequate in my opinion. There are a few important suggestions when it comes to structure, but the vast majority of my comments are related to format, grammar and clarity, and are ultimately to the discretion of the authors and the editing team.

Response: again, the authors thank the reviewer for such a positive response and taking the time to submit a thorough and much appreciated editorial review.

In regards to the payoffs of each tactic, the authors agree that they are speculative, in that we have no current way to measure the payoff (e.g. successful mating leading to a calf the following year). However, given these are mating tactics, it is not unreasonable to assume that the tactic should have some sort of payoff (otherwise why do it?) and given they are in breeding mode, the payoff must be related to breeding. To have the chance of breeding with a female, the male must join with the female, and we see this joining behaviour often. When comparing the breeding potential of males, those with a female should have a higher chance of breeding than those without – thus the joining with a female increases the breeding chance of that male. The statements have been softened as much as possible without excluding assumptions that we argue are not too much of a stretch.

The escorting of maternal females was not considered to be a separate tactic, but falls under the category of non-singing escorting (or singing whilst escorting). In our data, non-singing escorting (a maternal female) was included in the non-singing category throughout as stated in the methods. The non-singing category was referred to as non-singing physical competition, as this was often the outcome, whereas singing whilst escorting a maternal female was included in the singing category. To make this clearer, we have clarified this in that the physical competition category has been relabelled non-singing (including the phrase 'which often led to physical competition') and the singing category has been relabelled singing, clarifying that those categories include escorting maternal females (either singing or non-singing at the time of measurement). Also made it clearer that the non-singing tactic often involved physical competition but did not exclusively lead to this outcome.

Manuscript summary

An 18-year data set from the east coast of Australia, part of the migratory path for humpback whale breeding stock E1, examined trends in use of two mating tactics (physical competition vs. singing) as the population recovered from extensive exploitation. As a result of increasing population density, males transitioned from the singing strategy to physical competition.

Summary of review:

Overall, I think this is an outstanding article. This sort of long-term examination of humpback whale mating strategies is as needed as it is scarce. I commend the authors for the effort that so evidently supported the monitoring program for over 18 years. The methodology is adequate and the statistical analyses sufficient to answer the research questions. Most of the content within the sections flows very well, making this quite an enjoyable read. My main concern would be the assumptions regarding payoffs of each tactic, as I find them too speculative. The supplemental material was quite helpful in dissipating my concerns regarding sexing predictions. While I assume limiting to just two strategies (competition within mating groups and singing) better fits the data available, leaving the escorting of maternal females with little to no mention is inadequate in my opinion. There are a few important suggestions when it comes to structure, but the vast majority of my comments are related to format, grammar and clarity, and are ultimately to the discretion of the authors and the editing team.

Response: see above

Minor/complementary concerns

Response: each point has included a short response underneath.

1. Line 3: Consider separating the two ideas after the word issue. I assume the second sentence would start with "Recent studies..."

done

2. Line 10: The phrase "alternative mating strategy" makes it sound like singing and physical competition are the alternative to something else not mentioned. The easiest conclusion would be escorting of maternal females but that is barely mentioned. I would suggest to try

to improve clarity here.

Done. See response to maternal females above

3. Line 10: I strongly recommend against the use of personal pronouns but I rely on fellow reviewers and the editing team to provide additional input.

Response: I think the use of personal pronouns has increased in popularity. Though I was once against it and preferred the impersonal, I think using the personal does help with flow and readability. Also, where these pronouns are used is where we actually did the work which I think is justified.

4. Lines 10-12: I suggest re-arranging this as follows: "...as male density increased, the use of mating tactics shifted from singing to physical competition...".

Done

5. Line 17: Consider switching from "expressing" to "engaging in".

Done

6. Line 18: As of now, with the first mention of cost and benefit, it is a bit confusing. While greater detail is provided later on, the clarity of this sentence is limited. Additionally, while "easy" to consider in terms of methods, is it biologically relevant to limit success/cost to social affiliations? While logistically quite demanding, could be a good idea to complement with paternity data, not senseless given the magnitude of the data set.

Response: Re-written to clarify. Also clarified later on in the manuscript that a joining interaction does not necessarily mean a successful mating, but that by even joining with a female, the male has a better chance of mating with one (as compared to not joining a female). For any sort of paternity study, the population is about 30,000 whales. It would not be possible to assign calfs to fathers genetically as this would require genetic sampling of a large portion of a large population which would be almost impossible. In addition, we are (were) not permitted to sample calfs in this study.

7. Lines 19-21: This reads quite repetitive to the previous sentence.

These sentences have been re-written as per previous comment

8. Line 28: I recommend switching from "sneaker" to "sneaking".

"sneaker" in the literature, unchanged.

9. Lines 29-30: "Individuals have a set of rules" reads quite awkward. I recommend adopting the following structure: "...According to the fitness payoff (reproductive success), individuals should be able to decide when a specific tactic is advantageous (ref)...".

Response: Order of concepts changed but concept not changed to 'decide' as this suggests conscious thought in decision making, which, given insects also do it, is unlikely.

10. Line 44: Consider replacing the word "view" with "assess".

Done

11. Line 52: I suggest switching from “the cessation of whaling” to “the whaling moratorium”.

Done

12. Lines 56-57: Be mindful of repetitive expressions. I recommend the following: “The end of whaling led to an unprecedented recovery and an approximate 11% of annual growth (ref).”

Done

13. Lines 58-59: the sentence between commas is repetitive as this point was just mentioned. I urge the authors to replace it with “the surviving animals”.

Done

14. Lines 59-61: In order to increase clarity I suggest the following edit: “...Assuming humpback whales live for approximately 60 years, only now (2020’s) the population likely includes animals that have lived their entire lifespan...”.

done

15. Line 64: In order to increase clarity I suggest the following edit: “...Within breeding aggregations the operational sex-ratio is skewed towards males (ref)...”.

Done

16. Line 66: Add n to “alternative”.

Done

17. Lines 67-70: While the scenario where a female-male escort are then joined by additional males forming a competitive group, there are obviously many additional contexts that will produce the same result. For example, this leaves out the affiliation of additional animals (likely males) after the execution of surface-active behaviors. Therefore, I suggest the following edit: “The first is physical competition. Within competitive groups males physically fight...”.

Response: Changed to rephrase that this is one of the ways a competitive group is formed (albeit the most common way in our study site)

18. Line 76: Consider the following edit to improve flow: “...to attract females, mediate male-male competition, facilitate a male-lekking system...”.

Done

19. Line 79: I suggest switching from “then may” to “but may”.

Not done, the use of ‘then’ suggests a progression in time, which it is – lone then escorting

20. Line 95: As mentioned in the summary of the review, I am not convinced this is a successful reflection of the mating tactics payoff. Even throughout the manuscript the authors mention that main escorts can be easily replaced.

Response: We argue the statement in the main section “This does not assume a successful mating took place, but that the male had a greater chance of mating given it was escorting, and therefore had access to, a female” holds true.

Breeding behaviour, in our study area during this time of the migration, is defined by males attempting to join with females, sometimes using song, sometimes not, and sometimes attracting more males to join resulting in competitive behaviour. If we took a random sample of male whales, some that joined a female, some that didn't, we would assume the ones that joined had a greater chance of mating with a female than the ones that didn't. If there was the same proportion of males and females migrating over time, and singers were sighted joining available females more than non-singers in earlier years, with the opposite in later years, then we would assume by the same argument that singers had a better chance of mating with a female, than non-singers, in earlier years, and *vice versa*. The fact that male escorts can be replaced, as the reviewer points out, was accounted for in our study in that is a cost. The cost being that the original male does not have sole access to the female, it may be displaced, meaning its chances of mating with her are vastly reduced. We found in early studies (eluded to in the manuscript) that singers had a higher chance of this in higher male density conditions, likely because song is a loud broadcast signal that can attract other males.

We have clarified this assumption as much as possible throughout the manuscript by the use of the phrase ‘proxy for mating success’.

21. Line 97: Remove the comma after “37,000 whales”.

Done

22. Lines 110-119: This paragraph is either redundant and/or belongs in the introduction section.

Removed apart from the last part which defines our time period

23. Line 115: I suggest switching from “point” to “period”.

Done

24. Line 122: Consider replacing the word “singing” by “song cycle”.

Done

25. Line 124: I suggest switching from “allocated” to “determined to be” as genetic sexing is not based on assumptions to assess sex.

Response: Changed to assigned – not every male was biopsied as per the biopsy results, we used those results to make a prediction on the sex of the animals

26. Lines 125-131: This bit is redundant as it was mentioned within the methods.

Response: Agreed, it is difficult to write the results before the methods as it is difficult to understand the results without some context as to where the data came from. Removed the justification, kept the sentence outlining where the data came from.

27. Line 159: Remove the word “and” after “including”.

Done

28. Line 160: It makes more sense for me to refer to the grey area instead of line.

Done

29. Line 164: I suggest switching from a comma to colon after the word “timeline”.

Done

30. Line 169: Include the word “in” after “changes”.

Done

31. Line 176: It is not clear what the Smith et al., 2008 citation refers to.

Removed

32. Line 187: I suggest removing the connector (however) as the information that follows is not exactly contradicting but supporting.

Done

33. Line 190: Switch from “was” to “were”.

Done

34. Line 191: Remove the comma after “tactics”.

Not done, the commas act as brackets in this context

35. Line 212: See comment above graphics including 95% confidence interval.

Done

36. Line 236: The “for a period” looks a bit unfinished.

Changed to define a time

37. Lines 239-240: Consider the following edit to improve clarity: “...It was common for multiple singers to be present in the area at any one time...”

changed to may have been given it wasn't necessarily common

38. Lines 241-243: I suggest this: “...The start and end of song cycles depended on the number of males within a singer's social circle...”.

Not changed, singing can include multiple song cycles but the start and end of song was a definitive start singing and then stop singing (as a behaviour).

39. Lines 277-280: Possibly to consider for discussion but is it possible that 1) this could be a pattern observed only throughout migratory corridors and/or 2) acoustic pollution increased so that masking decreased the gain for singing displays?

Response to first point – difficult to say but why would whales carrying out breeding behaviour do something different during migration compared to on the breeding grounds – not beyond possibility but seems unlikely.

In **response to the second**, again unlikely. Song is loud and broadcasts well, masking can be an issue but only with close-by shipping. In our study area, there is background shipping noise but there little chance this noise masks singing. Even along the Australian coastline, shipping activity is relatively light compared to American coastlines and masking is not such an issue. Given that both the proximate study (hourly where noise levels would not have changed much) and population-study came to the same conclusions (that male density influences mating tactic choice) the evidence points towards this being a social effect rather than a noise effect.

40. Lines 284-285: Consider deleting the comma and the “if” after “2004”. Then, on the next sentence, I suggest the following edit: “...By 2014/2015, this proportion was reduced by half...”.

done

41. Lines 309-316: I consider this whole paragraph belongs in the discussion instead of results.

Not done – this provides a lead into the results in our opinion, otherwise the reader does not know where the data came from and the assumptions to consider when reading the results

42. Line 310: A group or a single female? Because I would assume the presumed benefits/costs are not the same in those two situations.

Response: changed to female noting we never had groups containing more than one female

43. Line 311: It still reads quite speculative for my taste.

Response: added in that this was considered to be a proxy for breeding success due to arguments outlined in response to the same point above

44. Lines 317-324: I consider this whole paragraph belongs in the methods instead of results.

Response: again, without context, the results will be difficult to interpret. Suggest leaving as in to help with readability.

45. Lines 321-323: Quite convoluted writing.

Changed

46. Line 331: See comment above: a group vs. a female.

Done

47. Line 347: Incorporate the word “daily” before “migrating”.

Done

48. Lines 394-396: This first sentence feels a bit too much like a copy/paste from the introduction.

changed

49. Lines 396-398: Consider the following edit: "...While rapid environmental change can be a strong selective force for individual-level behavioral plasticity, a fast population level response to such change can be beneficial in lowering its extinction risk (ref)..."

Done

50. Line 399: I would suggest to remove the word "such".

Done

51. Line 401-402: I recommend saying "...we have evidence that increasing local male density, and a severe change in population density, influenced the frequency..."

done

52. Line 405: Consider switching from "non-singing tactic" to "the physical competition tactic". Additionally, the two sentences within this line read quite disconnected.

Not done – as the reviewer points out the non-singer tactic is not just about competitive groups. The second point was addressed by reordering the sentences.

53. Line 414: Add "to" before "sing".

done

54. Lines 415-416: I suggest switching the comma after "female" and replace it with "or" and deleting the comma after "males" as well as the bit before the reference (for that female).

done

55. Line 419: Consider replacing the word "loud" with "easily widespread".

Not done – easily widespread suggests good propagation (not true, propagates just as well as any signal), loud refers to the source of the signal.

56. Lines 419-420: The writing here is a bit awkward. I suggest: "...The current study, as well as previous studies (refs), have documented a singing whale..."

done

57. Lines 425-426: Consider the following edit: "Whether being joined by another male while singing is a cost (e.g. male-male competition) or a benefit (e.g. male-male cooperation) is difficult to assess with..."

Done

58. Line 429. I recommend switching from "assume" to "suggest".

Done

59. Lines 430-431: Consider the following edit: "The benefit may be to cause the singer to stop and move out of the area, thus increasing the joiner's chances of associating with available females..."

done

60. Line 436: I suggest switching from "mating behaviors" to "strategies".

Not done – the strategy is the overall strategy, the behaviours form that strategy

61. Line 446: I suggest switching from “increases” to “grows” as right next to it there is “increasing”.

done

62. Line 450: Consider replacing the word “learned” as there is no evidence of learning from the current observations.

done

63. Lines 451-452: I recommend switching from “...low density of females and males...” to “...low population density...”.

done

64. Line 453: I recommend switching from “abundant” to “common”.

done

65. Lines 455-456: Consider the following edit: “...competitive groups were sighted more often, physical competition became the dominant tactic...”.

done

66. Line 457: I recommend switching from “expressed” to “has been recorded”.

Not done – this was stated by the authors of the cited paper

67. Line 463: Add an “a” after “successful”.

Not done as sentence changed due to comment from reviewer 1

68. Line 469: Was age determined somehow? The assumption that mature animals were present only in 2015 was confusing. Before that was mostly juveniles/subadultsthen?

Addressed in response to reviewer 1

69. Line 471: I recommend switching from “interesting” to “necessary”.

Not done – I don’t think it is necessary for the conclusions of this study, but would be an interesting next step

70. Line 477: I recommend switching from “these whales” to “humpback whales”.

Done

71. Line 480: See comment above regarding the two target tactics (singing vs. physical competition) being regarded as the only ones too reductionist.

Addressed in previous comments and changed to ‘non-singing’ which includes the category the reviewer eludes to.

72. Line 544: Singer location should (in my opinion) be referenced before when describing how changes in group structure surrounding singers occurred.

done

73. Line 571: I suggest labelling “base station” as “acoustic station” and “Emu Mountain” as “Theodolite station” to simplify the caption.

Not done- other activities occurred at base and calling EM by its name makes it easier to find on a map

74. Line 576: I do not see the point in specifying the color when it is all black.

Done

75. Lines 581-585: Consider the following edit: “...For this analysis, a specific period (2003-2004) was chosen as it had the most instances of identified singers and non-singing males. Within this timeframe, whales were migrating through the study area at a sufficiently low density to avoid confusion. After 2004, it became increasingly difficult to focal follow successfully...”.

done

76. I recommend switching from “singing” to “a song cycle”.

Not done as argued earlier

77. Line 598: A very brief description of how the social circle was defined could be useful in the caption of figure 6. Additionally, consider the following edit: “...was chosen as the most likely communication space...”.

done

78. Lines 603-604: consider the following edit: “...along with each group size and composition (singer, lone non-singing individual, mother-calf pair, mother-calf pair and escort(s), adult-only group) were recorded...”.

Not done- group size was not a measured factor as group composition included this

79. Line 611: Likely to be a single female but not always the case. See Félix & Novillo, 2015.

Done but noting our biopsy study found no examples of multiple female groups at our study site.

80. Line 627: I recommend including a brief statement on individual identification that allowed focal males to be considered independent samples.

done

81. Line 644: I suggest switching from “at a population level” to “at the population level”.

Done

82. Line 645: Consider the following edit: “...Daily observations, instead of individual records, were summarized...”.

Done

83. Line 647: Stating the duration (10 hours) again looks repetitive.

done

84. Line 649: I am not sure what the intended message is here.

reclarified

85. Line 650: Consider the following edit: "...These data were used to contextually sex individual whales as detailed in the supplementary results...".

reclarified

86. Lines 653-660: All this belongs in the results section.

done

87. Lines 661-663: I suggest include dependent/response variable between commas after "per day" and to remove "the response variable was the number of singers" on line 663.

Done

88. Line 669: I recommend separating into two sentences after "timeframe" starting the second sentence with "The latter was the smooth term...".

Done

89. Line 675: Consider eliminating "their".

done

90. Line 680: "To do this" sound a bit too informal of a tone for my taste.

Done

91. Line 685: I suggest switching from "of the subsequent" to "from subsequent".

Done

92. Line 699: A bit more details are needed when mentioning "acoustic data".

Done

93. Line 713: "tactic was either singing or non-singing behavior" reads repetitive.

Done

Reviewer #3 (Remarks to the Author):

The paper presents research from an extraordinary dataset. I don't think there would be a dataset comparable to this in terms of monitoring the recovery of a whale population. The idea behind the study is fantastic and ultimately this study should be published. However, there needs to be some clarity around some sections as they were confusing with the message that was coming across. Abstract could be restructured by bringing the sentence starting line 19 forward in the abstract, as this is pertinent information to the results that is presented before it. Also, +20000 should be >20000.

Response: the authors thank the reviewer for this positive response. The abstract was shortened to meet the journal's word limit and during this process the ordering of sentence was changed as per the reviewer's suggestion.

The first section of the results is where I struggled the most and needs to be re-written to be more concisely to improve the clarity. The last paragraph of this section provides a concise summary, which is the main thrust of this section, but that clarity is lost within the text of how it is currently written.

Response: improved clarify of first section of results as much as possible without losing important information. Shortened sentences and removed superfluous information and information that was included in the methods but not required for the understanding of the results.

Line 232 – “another singer may have started.” What is meant by “may?” This is an inaccurate term the way it is used. If a singer left the area and you can't prove another singer starts then delete this sentence.

Response: may was used in the context of 'sometimes', sometimes there was only one singer, sometimes another started up whilst this singer was still singer. Changed to clarify

Line 285 – Sentence starting “By 2014/2015...”. The graph suggests that 10 males per day equates to 0 singers not one singer.

Response: the graph intercept does not fall to zero. However, this sentence was changed due to a suggestion from reviewer 2.

Overall, I am struggling with the “in text” interpretation (lines 284-285) of figure 4. 2003/2004 10 males equated to two singers' figure 4 suggests 1 singer, not 2. Also, y-axis label on figure 4 is missing “day.”

Response: x axis fixed – box size issue. The reviewer is misinterpreting the graph and this should be addressed in the methods (which come after the results). The line is the relationship between singers and migrating males, which is a constant, the other two relationships are then compared to this relationship. Added a statement in the figure caption to clarify.

Figure 6 legend – females (black)? Should this be females (red).

Response: changed